 eLIFE

# Calmodulin-controlled spatial decoding of oscillatory Ca²⁺ signals by calcineurin

**Sohum Mehta[1], Nwe-Nwe Aye-Han[1], Ambhighainath Ganesan[2], Laurel Oldach[1], Kirill Gorshkov[1], Jin Zhang[1,3,4]\***

[1]Department of Pharmacology and Molecular Sciences, The Johns Hopkins University School of Medicine, Baltimore, United States; [2]Department of Biomedical Engineering, The Johns Hopkins University, Baltimore, United States; [3]The Solomon H. Snyder Department of Neuroscience, The Johns Hopkins University School of Medicine, Baltimore, United States; [4]Department of Oncology, The Johns Hopkins University School of Medicine, Baltimore, United States

**Abstract** Calcineurin is responsible for mediating a wide variety of cellular processes in response to dynamic calcium (Ca²⁺) signals, yet the precise mechanisms involved in the spatiotemporal control of calcineurin signaling are poorly understood. Here, we use genetically encoded fluorescent biosensors to directly probe the role of cytosolic Ca²⁺ oscillations in modulating calcineurin activity dynamics in insulin-secreting MIN6 β-cells. We show that Ca²⁺ oscillations induce distinct temporal patterns of calcineurin activity in the cytosol and plasma membrane vs at the ER and mitochondria in these cells. Furthermore, we found that these differential calcineurin activity patterns are determined by variations in the subcellular distribution of calmodulin (CaM), indicating that CaM plays an active role in shaping both the spatial and temporal aspects of calcineurin signaling. Together, our findings provide new insights into the mechanisms by which oscillatory signals are decoded to generate specific functional outputs within different cellular compartments.

**\*For correspondence:**
jzhang32@jhmi.edu

**Competing interests:** The authors declare that no competing interests exist.

## Introduction

Calcium (Ca²⁺) is a ubiquitous and universal intracellular signal whose remarkable biological versatility is a product of diverse patterns of spatial and temporal regulation (**Thomas et al., 1996**; **Berridge et al., 2000**; **Dupont et al., 2011**), most notably in the form of repetitive, transient elevations in cytosolic Ca²⁺ concentrations, or Ca²⁺ oscillations. In general, oscillatory signals are thought to function as a critical biological regulator by allowing a single message to encode multiple types of information through variations in the frequency, amplitude, and spatial characteristics of the signal (**Cheong and Levchenko, 2010**; **Ganesan and Zhang, 2012**), thereby promoting specificity in the regulation of downstream targets. Ca²⁺ oscillations in particular are known to regulate numerous processes including gene expression (**Negulescu et al., 1994**; **Lewis, 2003**), exocytosis (**Tse et al., 1993**; **Pasti et al., 2001**), and excitation-contraction coupling (**Viatchenko-Karpinski et al., 1999**; **Maltsev and Lakatta, 2007**), and Ca²⁺ oscillations have been shown to significantly enhance the specificity and efficiency of Ca²⁺-regulated processes (**Dolmetsch et al., 1998**; **Li et al., 1998**; **Kupzig et al., 2005**). Cells are primarily dependent on a single effector protein, the Ca²⁺ sensor calmodulin (CaM), to transduce Ca²⁺ signals. CaM sits at the epicenter of Ca²⁺ signaling, modulating the activity of a vast array of target proteins throughout the cell (**Persechini and Stemmer, 2002**). CaM is also thought to play a prominent role in the decoding of Ca²⁺ oscillations, largely via the differential activation of target proteins such as the Ca²⁺- and CaM-dependent phosphatase calcineurin (**Saucerman and Bers, 2008**; **Song et al., 2008**; **Parekh, 2011**; **Slavov et al., 2013**).

One of the major targets of CaM in almost all eukaryotic cells (**Hilioti and Cunningham, 2003**), calcineurin is involved in regulating diverse physiological processes, including cell proliferation,

**eLife digest** Cells need to be able to communicate with other cells, and they employ a variety of molecules and ions to send messages to each other. When calcium ions are used for these communications, the concentration of the ions typically rises and falls in a wave-like pattern. The size and shape of these 'calcium waves' contains information that is needed by organs as diverse as the heart and the brain.

Most cells detect calcium waves using a sensor molecule called calmodulin. This, in turn, activates an enzyme called calcineurin. However, relatively little is known about the ways in which calcium waves shape the activity of calcineurin, even though calcium signaling is very common.

Mehta et al. have now clarified this relationship by studying how calcium ions affect the activity of calcineurin molecules inside pancreatic cells. The response of calcineurin to calcium depends on position inside the cell. In the cytosol and at the plasma membrane that encloses the cell, calcium waves trigger a very fast 'step-like' increase in calcineurin activity. By contrast, at the surface of certain organelles within the cell, the calcium waves cause the calcineurin activity to rise and fall in a wave-like pattern.

Experiments designed to identify the molecular mechanism behind this difference revealed that the answer lies in the distribution of calmodulin, the intermediate between calcium and calcineurin. At the surface of organelles, there is less calmodulin available to activate calcineurin than in the cytosol or at the plasma membrane. As a result, calcineurin activity in the vicinity of organelles is vulnerable to being canceled out by the actions of other enzymes. When more calmodulin is available, this canceling out does not occur, which is how wave-like input can lead to step-like output.

By identifying the mechanism by which a single signal—a calcium wave—generates distinct responses in the same target molecule—calcineurin—depending on subcellular location, Mehta et al. have identified a process that is relevant to a wide range of biological systems.

differentiation, and death, as well as gene expression, secretion, immune function, learning, and memory (reviewed in *Rusnak and Mertz, 2000*; *Aramburu et al., 2004*). However, the precise spatiotemporal regulation of calcineurin signaling remains poorly understood. $Ca^{2+}$ oscillations have previously been shown to enhance calcineurin-mediated transcriptional regulation (*Dolmetsch et al., 1998*; *Li et al., 1998*; *Tomida et al., 2003*; *Wu et al., 2012*), and studies have also shown that the $Ca^{2+}$ oscillatory frequency is a critical determinant of calcineurin-dependent hypertrophic signaling in cardiomyocytes (*Colella et al., 2008*; *Saucerman and Bers, 2008*) and long-term depression in neurons (*Mehta and Zhang, 2010*; *Li et al., 2012b*; *Fujii et al., 2013*). Nevertheless, the precise relationship between $Ca^{2+}$ oscillations and calcineurin signaling has yet to be elucidated. Similarly, calcineurin dephosphorylates multiple target proteins located throughout the cell (*Cameron et al., 1995*; *Wang et al., 1999*; *Bandyopadhyay et al., 2000*; *Cereghetti et al., 2008*; *Tandan et al., 2009*; *Bollo et al., 2010*), and although spatial compartmentalization is suspected to play an important role in regulating calcineurin signaling (*Heineke and Ritter, 2012*), this phenomenon has yet to be directly examined.

Both $Ca^{2+}$ oscillations and calcineurin signaling are known to play important roles in pancreatic β-cells. $Ca^{2+}$ oscillations are responsible for driving the pulsatile secretion of insulin (*Hellman et al., 1992*; *Tengholm and Gylfe, 2008*), and the chronic inhibition of calcineurin, which is a common form of immunosuppressive therapy, is often accompanied by the onset of diabetes (*Heisel et al., 2004*; *Chakkera and Mandarino, 2013*). In the present study, we use a variety of genetically encoded fluorescent reporters to directly investigate the spatiotemporal dynamics of calcineurin signaling in response to cytosolic $Ca^{2+}$ oscillations in MIN6 β-cells. We were able to observe distinct subcellular patterns of calcineurin activity in the cytosol and plasma membrane, where calcineurin appeared to integrate the oscillatory input, vs at the ER and mitochondrial surfaces, where calcineurin activity was observed to oscillate. Furthermore, an exploration of the molecular determinants involved in regulating calcineurin activity revealed that significant differences in the subcellular distribution of free $Ca^{2+}$/CaM are responsible for generating these discrete activity patterns. Our findings provide the first evidence that oscillatory signals are capable of differentially regulating calcineurin activity and suggest a more active role for CaM in transducing oscillatory $Ca^{2+}$ signals.

## Results

### Distinct zones of subcellular calcineurin activity in MIN6 cells

To investigate the spatiotemporal regulation of calcineurin signaling in response to $Ca^{2+}$ oscillations in β-cells, we engineered an improved version of our previously described FRET-based calcineurin activity reporter (CaNAR) (**Newman and Zhang, 2008**) by optimizing the donor and acceptor fluorescent protein pair (**Figure 1**). We then targeted this reporter, called CaNAR2, to the cytosol (cytoCaNAR2), plasma membrane (pmCaNAR2), mitochondrial outer membrane (mitoCaNAR2), and ER surface (erCaNAR2) via in-frame fusion with a C-terminal nuclear export signal (NES) (**Ullman et al., 1997**), an N-terminal motif derived from Lyn kinase (Lyn) (**Gao and Zhang, 2008**; **Depry et al., 2011**), an N-terminal motif derived from DAKAP1a (DAKAP) (**DiPilato et al., 2004**), and an N-terminal motif derived from cytochrome P450 (CYP450) (**Szczesna-Skorupa et al., 1998**), respectively (**Figure 2A–E**). Each targeted CaNAR2 variant was co-expressed in MIN6 β-cells along with a diffusible version of the genetically encoded, red-fluorescent $Ca^{2+}$ indicator RCaMP (**Akerboom et al., 2013**), allowing us to simultaneously visualize and characterize the coordination of subcellular calcineurin activity with cytosolic $Ca^{2+}$ levels. Membrane depolarization induced by tetraethylammonium chloride (TEA) treatment produced robust oscillations in RCaMP fluorescence intensity, which were consistent with the cytosolic $Ca^{2+}$ oscillations previously observed in MIN6 cells (**Landa et al., 2005**; **Ni et al., 2010**) (**Figure 2F–I**, red curves).

Using both cytosolic and plasma membrane-targeted CaNAR2, we were able to observe integrating, step-like patterns of calcineurin activity in response to TEA-induced $Ca^{2+}$ oscillations, with each step-increase in calcineurin activity synchronized to a cytosolic $Ca^{2+}$ peak, as measured by RCaMP (**Figure 2F,G**, **Figure 2—figure supplement 1A,B**). We have seen previously that the FRET signal from CaNAR, which is based on calcineurin-dependent dephosphorylation of the N-terminal domain of nuclear-factor of activated T-cells (NFAT), is not easily reversed and can remain elevated even after cytosolic $Ca^{2+}$ has returned to basal levels (**Newman and Zhang, 2008**). Indeed, these observations appear to confirm previous reports suggesting that calcineurin and NFAT behave as signal integrators to form a working memory of oscillatory $Ca^{2+}$ signals, owing to the rapid dephosphorylation and slow rephosphorylation kinetics of NFAT family members (**Tomida et al., 2003**; **Colella et al., 2008**). To our surprise, however, mitochondrial and ER-targeted CaNAR2 both exhibited far more reversible responses to cytosolic $Ca^{2+}$ oscillations (**Figure 2H,I**, **Figure 2—figure supplement 1D,E**). This difference was particularly striking for erCaNAR2, which exhibited almost perfect calcineurin activity oscillations upon TEA stimulation. Furthermore, these response patterns were independent of the CaNAR expression level (**Figure 2—figure supplement 1**), indicating that they reflect genuine differences in endogenous calcineurin signaling.

### PKA opposes localized calcineurin signaling at the ER

Our findings suggest the existence of two discrete subcellular zones (e.g., cytosol/plasma membrane and ER/mitochondria) with distinct calcineurin signaling activities in MIN6 β-cells. We therefore investigated the molecular mechanisms responsible for defining these differential signaling zones. In particular, we focused on the reversibility of the CaNAR response at the ER surface, which suggested that calcineurin activity within this region of the cell is being precisely balanced by the action of endogenous kinases. A number of kinases have been shown to re-phosphorylate NFAT (reviewed in **Crabtree and Olson, 2002**; **Hogan et al., 2003**) and are thus potentially capable of reversing the response from CaNAR. Notably, PKA phosphorylates multiple residues in NFAT (**Beals et al., 1997**; **Sheridan et al., 2002**), and PKA has repeatedly been shown to antagonize calcineurin signaling in the regulation of a variety of cellular processes such as exocytosis (**Lester et al., 2001**), excitation-contraction coupling (**Santana et al., 2002**), and mitochondrial division (**Cribbs and Strack, 2007**). We therefore tested whether PKA activity is involved in reversing the ER-localized CaNAR response. In addition, we also tested the role of the $Ca^{2+}$- and diacylglycerol-stimulated protein kinase (PKC), which may also phosphorylate NFAT (**San-Antonio et al., 2002**).

Blocking PKA activity by treating TEA-stimulated MIN6 cells with the PKA inhibitor H89 appeared to abolish erCaNAR2 oscillations, leading to a delayed but pronounced increase in the CaNAR response (**Figure 3A,B**). In addition, H89 treatment completely abolished the $Ca^{2+}$ oscillations, which is consistent with the behavior of a previously described $Ca^{2+}$/cAMP/PKA oscillatory circuit in MIN6 β-cells (**Ni et al., 2010**). Curiously, the increased response from erCaNAR2 coincided with the

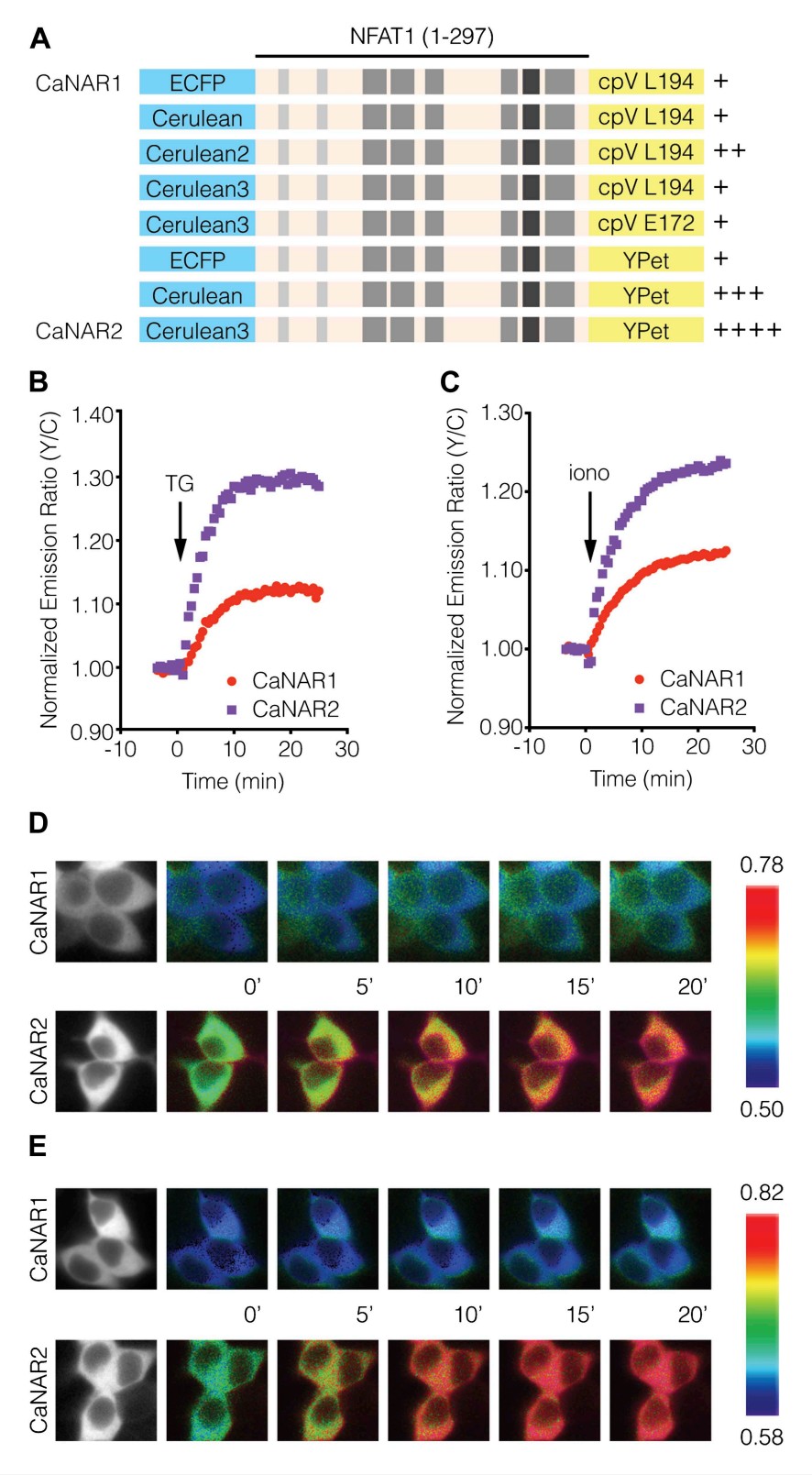

**Figure 1**. Development and characterization of CaNAR2. (**A**) Schematic depicting the CaNAR variants tested. FRET pair optimization was performed by replacing the original ECFP and circularly permuted Venus (cpV) L194 of CaNAR1 (top) with Cerulean, Cerulean2, Cerulean3, cpV E172, or YPet. The responses from each construct are

*Figure 1. Continued on next page*

*Figure 1. Continued*

indicated as follows: +, ~5–10%; ++, ~10–15%; +++, ~15–20%; ++++, >20%. (**B** and **C**) Representative time-courses comparing the yellow/cyan (Y/C) emission ratio changes from CaNAR1 and CaNAR2 in HEK293 cells treated with (**B**) 1 µM thapsigargin (TG) or (**C**) 1 µM ionomycin (iono). CaNAR2 exhibits an at least twofold greater response in each condition. (**D** and **E**) Pseudocolored images showing the responses of CaNAR1 and CaNAR2 to (**D**) 1 µM TG or (**E**) 1 µM iono in HEK293 cells. Warmer colors correspond to higher FRET ratios. Cyan fluorescence images (left) show the cellular distribution of CaNAR1 and CaNAR2 fluorescence.

attenuation of $Ca^{2+}$ oscillations and the return of $Ca^{2+}$ to basal levels. Despite the apparent lack of $Ca^{2+}$ signaling, the observed increase in the erCaNAR2 response was nonetheless specifically caused by calcineurin activity, as the increase could be blocked by addition of the calcineurin inhibitor cyclosporin A prior to H89 treatment (*Figure 3E*). Moreover, addition of the PKC inhibitor Gö6983 to TEA-stimulated MIN6 cells also resulted in a steadily increasing erCaNAR2 response, accompanied by a transition from $Ca^{2+}$ oscillations to persistently elevated cytosolic $Ca^{2+}$ levels (*Figure 3C,D*).

The fact that both inhibitor treatments altered the underlying $Ca^{2+}$ dynamics prevented us from reaching any firm conclusions regarding the roles of PKA and PKC based on these experiments. We therefore sought an alternative method for generating repetitive $Ca^{2+}$ spikes in MIN6 cells. TEA functions by blocking plasma membrane $K^+$ channels, leading to membrane depolarization, and thereby activating voltage-gated $Ca^{2+}$ channels (*Hille, 1967*; *Wang and Greer, 1995*). The direct addition of KCl also promotes depolarization-induced $Ca^{2+}$ influx in electrically excitable cells such as β-cells (*Bianchi and Shanes, 1959*; *Powell et al., 1984*; *Bading et al., 1993*; *Graef et al., 1999*; *Macías et al., 2001*; *Everett and Cooper, 2013*), and repeated cycles of KCl addition and wash-out were successfully able to mimic TEA-stimulated oscillations in cytosolic $Ca^{2+}$ levels and ER-localized calcineurin activity in MIN6 cells (*Figure 4A*).

Using this approach, we found that the addition of H89 resulted in an integrative, step-like response from erCaNAR2 (*Figure 4B*), much like that seen with cytoCaNAR2 in TEA-stimulated cells (*Figure 2F*), although RCaMP did not appear to respond under these conditions. However, we were able to confirm the generation of $Ca^{2+}$ spikes using a higher affinity probe, YC-Nano50 (*Figure 4—figure supplement 1*; *Horikawa et al., 2010*). In contrast to H89 treatment, the inclusion of Gö6983 to inhibit PKC activity did not appear to alter the response of erCaNAR2 to successive KCl treatments (*Figure 4C*). Here, the addition of the inhibitor resulted in an immediate increase in the erCaNAR2 FRET signal, though this was most likely due to non-specific increases in the background fluorescence due to the addition of Gö6983, which is fluorescent. Nevertheless, KCl-induced erCaNAR2 oscillations were still clearly observable above this increased background, as were the RCaMP $Ca^{2+}$ spikes.

Our findings indicated that PKA activity, but not PKC activity, antagonizes calcineurin near the ER surface, thereby giving rise to ER-localized calcineurin activity oscillations in MIN6 cells. Based on these results, we tested whether differences in the level of PKA activity present in the cytosol and ER might contribute to differential calcineurin response patterns. To do so, we utilized variants of the FRET-based PKA activity reporter AKAR4 (*Depry et al., 2011*) that were localized to the cytoplasm or ER via fusion to the targeting sequences described above (*Figure 5A*). We then compared the relative amounts of PKA activity in these two compartments by normalizing the TEA-stimulated PKA responses (*Figure 5C,D*) with respect to the total amount of PKA activity available in the cytosol and ER, which was defined as the maximum subcellular response observed upon combined treatment with the adenylyl cyclase activator forskolin (Fsk) and the general phosphodiesterase inhibitor 3-isobutyl-1-methylxanthine (IBMX). Interestingly, we detected slightly less PKA activity at the ER, with TEA-stimulated AKAR4 responses reaching 55.4 ± 5.9% and 39.8 ± 3.3% (p = 0.0249) of the maximum dynamic range in the cytosol and at the ER, respectively (*Figure 5B*). We also found that setting PKA activity to maximum levels using IBMX pretreatment in conjunction with KCl stimulation, thus clamping PKA activity in these regions (*Figure 5—figure supplement 1*), did not affect the subcellular CaNAR response patterns (*Figure 5E–H*).

## Calcineurin activation kinetics are uniformly synchronized with cytosolic $Ca^{2+}$ oscillations

Given that $Ca^{2+}$ oscillations actually induce a lower amount of PKA activity at the ER surface compared with the cytosol, the apparent inability of PKA to reverse the cytosolic CaNAR response is puzzling. One possible explanation is that the kinetics of calcineurin activation itself may in fact differ in these

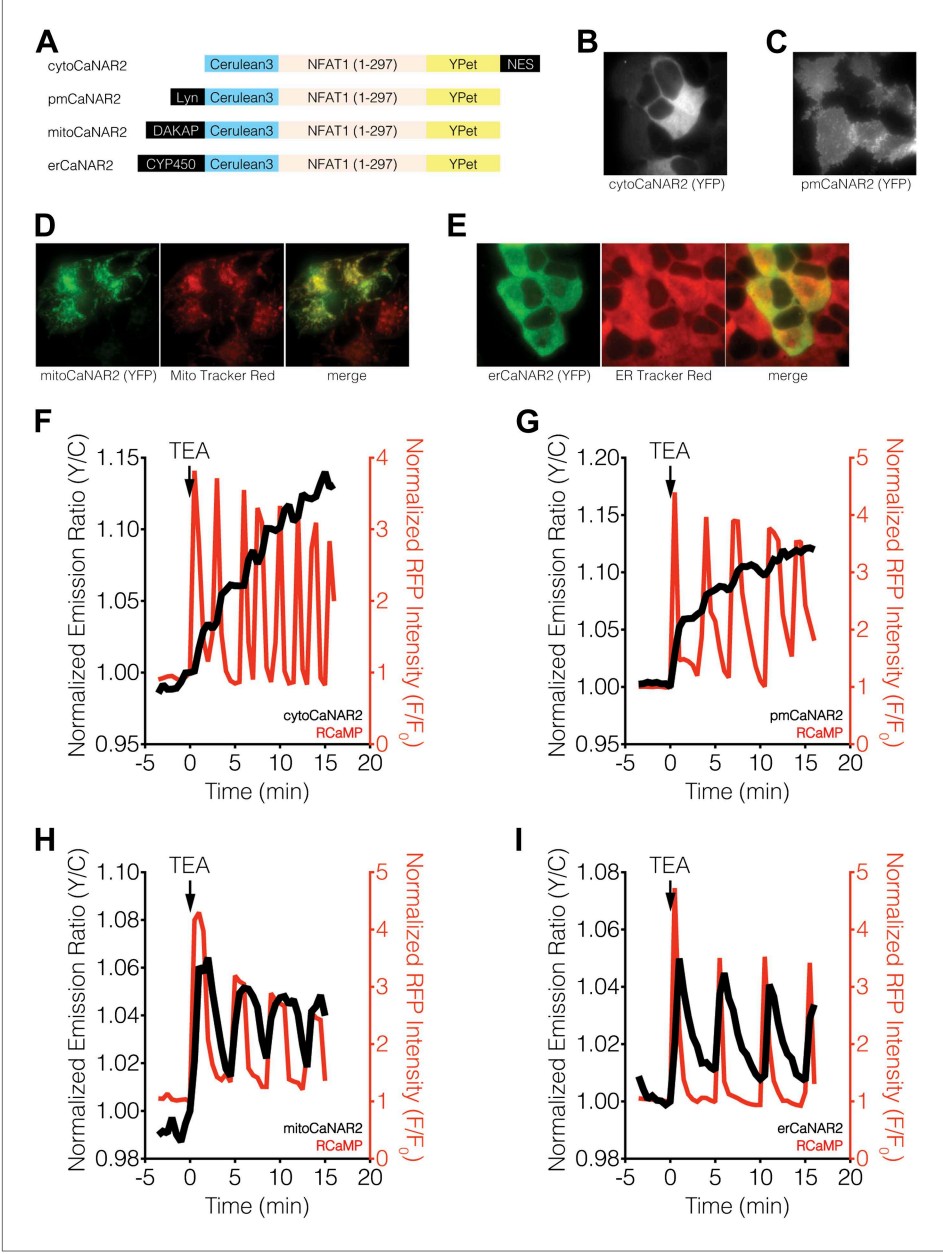

**Figure 2**. Subcellular calcineurin activity dynamics in response to $Ca^{2+}$ oscillations in MIN6 cells. (**A**) Schematic illustrating the domain structures of the subcellularly targeted variants of the CaNAR2 biosensor. (**B** and **C**) Yellow fluorescence images showing the biosensor distribution in transiently transfected MIN6 cells expressing (**B**) cytoCaNAR2 and (**C**) pmCaNAR2. (**D** and **E**) Fluorescence images showing the localization of mitoCaNAR2 and erCaNAR2. MIN6 cells expressing (**D**) mitoCaNAR2 or (**E**) erCaNAR2 were stained with MitoTracker Red or ER-Tracker Red, respectively. Image series corresponds to biosensor fluorescence (YFP, left), dye fluorescence (middle), and merged (right). (**F**–**I**) Representative time-courses showing the yellow/cyan (Y/C) emission ratio changes from (**F**) cytoCaNAR2 (n = 29), (**G**) pmCaNAR2 (n = 22), (**H**) mitoCaNAR2 (n = 11), and (**I**) erCaNAR2 (n = 15) (black curves), along with the red fluorescence intensity changes from RCaMP (red curves), in MIN6 cells stimulated with 20 mM TEA.

The following figure supplement is available for figure 2:

**Figure supplement 1**. CaNAR expression levels do not affect subcellular calcineurin response dynamics.

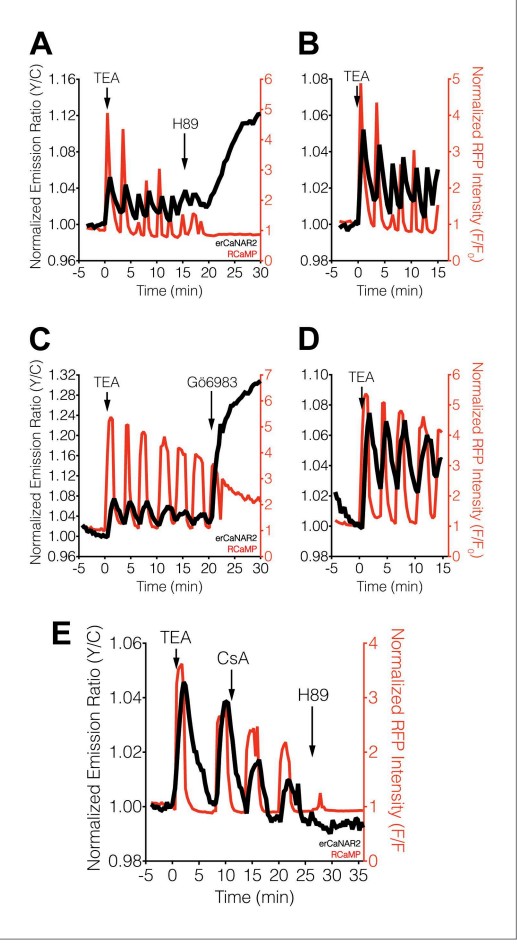

**Figure 3**. Effect of PKA and PKC inhibition on TEA-induced ER calcineurin activity oscillations. (**A**) Representative time-course showing the effect of 20 µM H89 treatment on the TEA-stimulated responses from erCaNAR2 (black curve) and RCaMP (red curve) in MIN6 cells (n = 11). (**B**) Expanded time-course showing the TEA-stimulated responses from (**A**). (**C**) Representative time-course showing the effect of 10 µM Gö6983 on the TEA-stimulated responses from erCaNAR2 (black curves) and RCaMP (red curve) in MIN6 cells (n = 6). (**D**) Expanded time-course showing the TEA-stimulated responses from (**C**). (**E**) Representative time-course showing the responses from erCaNAR2 (black curve) and RCaMP (red curve) in MIN6 cells treated with 20 mM TEA, 6 µM cyclosporin A (CsA), and 20 µM H89 at the indicated times (n = 3). DOI: 10.7554/eLife.03765.006

two subcellular zones. Specifically, we reasoned that calcineurin may remain in an activated state for longer periods in the cytosol than at the ER surface in response to the individual $Ca^{2+}$ pulses that occur during $Ca^{2+}$ oscillations, thus potentially rendering cytosolic calcineurin activity less prone to the antagonistic effects of cytosolic PKA activity. To test this hypothesis, we took advantage of the intrinsic conformational changes that are associated with calcineurin activation.

Calcineurin exists as a stable heterodimer between a regulatory subunit (CNB) and a catalytic subunit (CNA) and becomes activated when $Ca^{2+}$-bound CaM ($Ca^{2+}$/CaM) binds to the regulatory arm of CNA, thus driving a conformational change that removes calcineurin from its basal, auto-inhibited state (*Rusnak and Mertz, 2000*; *Wang et al., 2008*; *Rumi-Masante et al., 2012*). Guided by this information, we constructed a FRET-based reporter for monitoring calcineurin activation by sandwiching the catalytic subunit of calcineurin between Cerulean (i.e., CFP) and Venus (i.e., YFP) (*Figure 6A*). The resulting Calcineurin Activation Ratiometric indicator, or CaNARi, exhibited a robust increase in the cyan-to-yellow fluorescence emission ratio, which tracked closely with the increase in $Ca^{2+}$ detected using RCaMP, in response to cytosolic $Ca^{2+}$ elevation in HEK293 cells (*Figure 6B*). This $Ca^{2+}$-induced FRET change could be blocked by pretreatment with the membrane-permeable CaM antagonist W7, indicating that the CaNARi response is in fact dependent on the binding of $Ca^{2+}$/CaM (*Figure 6C*).

We then tested whether the kinetics of calcineurin activation display subcellular variations in response to $Ca^{2+}$ oscillations by localizing CaNARi to the cytosol and plasma membrane, as well as to the mitochondrial and ER surfaces, in MIN6 cells via fusion to the aforementioned targeting sequences (*Figure 7A*). As above, TEA stimulation induced cytosolic $Ca^{2+}$ oscillations in the CaNARi-expressing MIN6 cells, as determined using co-expressed RCaMP. However, in contrast to the subcellular CaNAR responses, the responses from the targeted CaNARi variants were similar in all of the subcellular compartments that we examined (*Figure 7C–F*). Specifically, the responses from CaNARi in each subcellular region closely matched the cytosolic $Ca^{2+}$ dynamics, rapidly increasing as $Ca^{2+}$ levels rose and subsequently decreasing as $Ca^{2+}$ fell back to basal levels, and revealed no major subcellular differences in how quickly calcineurin was turned on or off in response to oscillating $Ca^{2+}$ levels. These results indicate that the kinetics of calcineurin activation are similar throughout the cell under these conditions.

## Subcellular differences in $Ca^{2+}$/CaM levels regulate calcineurin activity dynamics

Although we were unable to observe any subcellular differences in the kinetics of calcineurin activation using CaNARi, our findings do not completely rule out a possible role for variations in subcellular

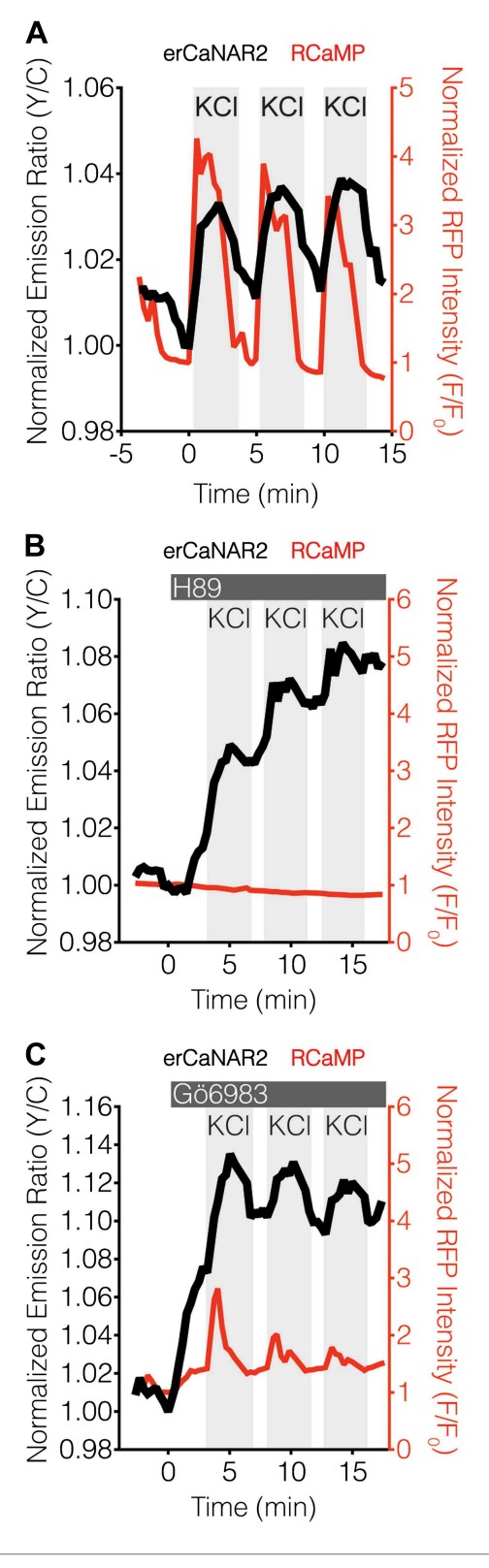

**Figure 4**. PKA antagonizes ER calcineurin activity in MIN6 cells. (**A**) Manual induction of Ca²⁺ oscillations in MIN6 cells via repeated addition and washout of *Figure 4. Continued on next page*

calcineurin activation in regulating subcellular calcineurin activity. In fact, a closer inspection of the subcellularly targeted CaNARi responses revealed that both mitochondrial and ER-targeted CaNARi exhibited slightly smaller TEA-stimulated responses compared with the cytosolic and plasma membrane probes (*Figure 7B*). We also observed no correlation between the CaNARi response amplitude and the biosensor expression level (*Figure 7—figure supplement 1*), suggesting this difference was not an artifact of reporter expression. Although the trend was not statistically significant, these results lend credence to the hypothesis that different amounts of calcineurin are being activated in different parts of the cell.

We then investigated whether variations in upstream signaling components could be generating differences in subcellular calcineurin activation. Using subcellularly targeted versions of the green-fluorescent Ca²⁺ sensor GCaMP3 (*Tian et al., 2009*), we were unable to detect any obvious differences in TEA-stimulated Ca²⁺ dynamics in the cytosol or at the ER surface with respect to the RCaMP response, although we did observe a steady rise in the basal Ca²⁺ level using ER-targeted GCaMP3 (*Figure 8*). These results largely agree with our subcellular CaNARi data and suggest that the divergent subcellular calcineurin activity patterns are not caused by local differences in the underlying Ca²⁺ dynamics (e.g., influx or efflux). In addition, only a small difference was observed when we compared the magnitude of these local Ca²⁺ signals, with a slightly lower amount of Ca²⁺ near the ER surface than in the cytosol (71.4% of max Ca²⁺ in the cytosol vs 61.2% at the ER, p = 0.0049).

Calcineurin activation is also completely dependent on the binding of Ca²⁺/CaM, and multiple studies have shown that CaM is a limited cellular resource (reviewed in *Persechini and Stemmer, 2002*; *Saucerman and Bers, 2012*). Indeed, the concentration of free CaM often reaches only 75 nM (*Persechini and Stemmer, 2002*; *Wu and Bers, 2007*; *Saucerman and Bers, 2012*), whereas CaM targets can be in excess of 20 μM (*Saucerman and Bers, 2012*). Similarly, based on the kinetics of Ca²⁺ binding to and dissociation from the N- and C-lobes of CaM, it was calculated that Ca²⁺/CaM would only diffuse ~0.1 μm before Ca²⁺ begins to dissociate, suggesting that Ca²⁺/CaM primarily acts as a highly localized signal (*Saucerman and Bers, 2012*). Rapid, long-range signaling may even require actively transporting CaM to other parts of the cell (*Deisseroth et al., 1998*). Thus, taking into account both the

*Figure 4. Continued*

15 mM KCl. Addition of KCl rapidly increases the responses from RCaMP (red curve) and erCaNAR2 (black curve), which are then both reversed upon washout. Repeating this process generates oscillatory responses. (**B**) Representative time-course showing the effect of 20 μM H89 treatment on the KCl-induced erCaNAR2 (black curve) and RCaMP (red curve) response in MIN6 cells (n = 5). H89 was added prior to the initial KCl treatment, and the H89 concentration in the experiment was maintained by re-addition of 20 μM H89 after washing out KCl. Although RCaMP did not detect $Ca^{2+}$ responses in this experiment, $Ca^{2+}$ spikes could clearly be seen using a high-affinity $Ca^{2+}$ probe (*Figure 4—figure supplement 1*). (**C**) Representative time-course showing the effect of 10 μM Gö6983 treatment on the KCl-induced erCaNAR2 (black curve) and RCaMP (red curve) response in MIN6 cells (n = 4). Gö6983 was added prior to the initial KCl treatment, and the Gö6983 concentration in the experiment was maintained by re-addition of 10 μM Gö6983 after washing out KCl.

The following figure supplement is available for figure 4:

**Figure supplement 1**. YC-Nano50 detects KCl-induced $Ca^{2+}$ influx in the presence of H89.

scarcity of $Ca^{2+}$/CaM and its limited range of action, along with the fact that calcineurin activity increases as a direct function of CaM concentration (*Quintana et al., 2005*), it is conceivable that even subtle variations in the availability of $Ca^{2+}$/CaM could have a significant effect on local calcineurin activity (*Figure 9A*).

To investigate subcellular differences in $Ca^{2+}$/CaM levels, we used the FRET-based biosensor BSCaM-2, which has been used previously to measure free $Ca^{2+}$/CaM levels in living cells (*Persechini and Cronk, 1999*; *Tran et al., 2003*), as well as to investigate subcellular differences in free $Ca^{2+}$/CaM levels (*Teruel et al., 2000*). This reporter features a modified version of the $Ca^{2+}$/CaM-binding sequence from avian smooth muscle myosin light chain kinase that binds $Ca^{2+}$/CaM with a $K_d$ of 2 nM. Calcineurin has been shown to bind $Ca^{2+}$/CaM with sub-nanomolar affinity (*Hubbard and Klee, 1987*; *Quintana et al., 2005*), therefore we reasoned that BSCaM-2 should offer a fair approximation of how much $Ca^{2+}$/CaM is locally accessible by calcineurin. Our model predicts that free $Ca^{2+}$/CaM is less abundant in the vicinity of the ER compared with the bulk cytosol (*Figure 9A*, left), and indeed we observed a significantly lower (p < 0.0001) FRET response when BSCaM-2 was tethered to the ER surface than when it was able

to diffuse freely through the cytosol (*Figure 9B*), suggesting that less $Ca^{2+}$/CaM is present near the ER. These responses are similar to those observed by Teruel and colleagues, who reported smaller increases in free nuclear $Ca^{2+}$/CaM levels compared with the cytosol in response to $Ca^{2+}$ transients (*Teruel et al., 2000*). Our results also appear to reflect actual differences in subcellular $Ca^{2+}$/CaM levels as opposed to $Ca^{2+}$/CaM buffering, given that biosensor expression levels did not appear to affect the FRET responses (*Figure 9—figure supplement 1*). More importantly, we found that overexpressing mCherry-tagged CaM was able to rescue most of the difference between the ER and cytosolic FRET responses (*Figure 9B*, 'ER+CaM').

Our model suggests that lower levels of $Ca^{2+}$/CaM will result in weaker calcineurin activation near the ER surface (*Figure 9A*, red and green curves), which would in turn translate into lower levels of calcineurin activity that are more susceptible to antagonism by PKA activity. To test this model directly, we reasoned that if calcineurin activity is in fact being affected by local $Ca^{2+}$/CaM levels, it should then be possible to generate cytosol-like CaNAR responses with erCaNAR2 by overexpressing CaM. Remarkably, combining mCherry-tagged CaM overexpression with the application of repeated, KCl-induced $Ca^{2+}$ transients in CaNAR-expressing cells reveals that this is indeed the case. In contrast to cells expressing erCaNAR2 alone, which exhibit oscillatory FRET responses in response to repeated KCl stimulation and washout (*Figure 9C*), the co-expression of CaM-mCherry alongside erCaNAR2 clearly results in integrated calcineurin activity responses similar to those seen in the cytosol (*Figure 9D*). Conversely, reducing the amount of available $Ca^{2+}$/CaM should lead to ER-like CaNAR oscillations in the cytosol. Indeed, pretreating cells with a low dose (20 μM) of the CaM antagonist W7 gave rise to oscillatory responses from cytoCaNAR2, in contrast to integrating responses in cells lacking W7 pretreatment (*Figure 9E,F*). Taken together, our results strongly suggest that free concentrations of $Ca^{2+}$/CaM are limiting near the ER surface and thus significantly modulate the local, $Ca^{2+}$ oscillation-induced calcineurin activity dynamics in this subcellular region.

## Discussion

The spatiotemporal regulation of calcineurin signaling has come under increased scrutiny of late. Recently, calcineurin responses in cortical neurons treated with the amyloid-β peptide were shown to

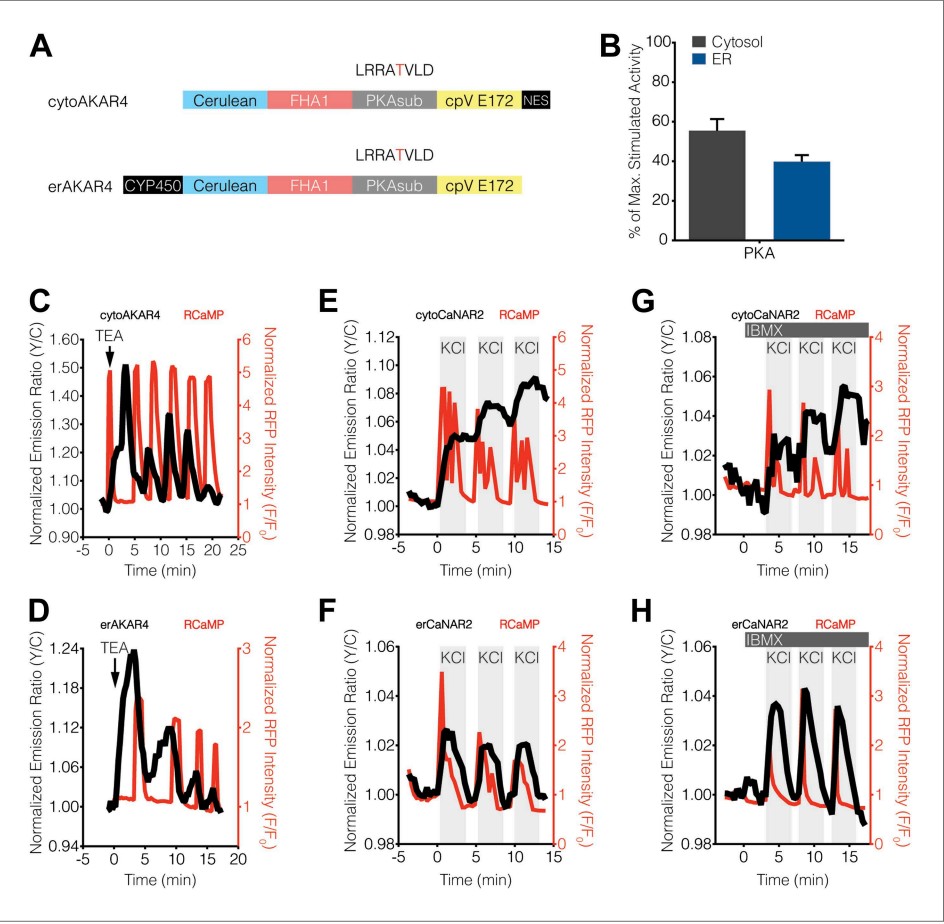

**Figure 5**. Characterization of cytosolic and ER-localized PKA activity in MIN6 cells. (**A**) Schematic illustrating the domain structures of cyto- and erAKAR4. (**B**) Comparison of fractional PKA activity levels in the cytosol and ER of MIN6 cells. To determine the relative fraction of PKA activity being induced by TEA stimulation in each subcellular compartment, the FRET ratio change from each TEA-induced cytoAKAR4 or erAKAR4 response was divided by the maximum FRET ratio change observed upon co-stimulation with 50 µM Fsk and 100 µM IBMX. Data shown are presented as mean ± SEM, with n = 36 and 34 for cytoAKAR4 and erAKAR4, respectively. (**C** and **D**) Representative time-courses showing the responses from (**C**) cytoAKAR4 and (**D**) erAKAR4 (black curves), along with RCaMP (red curves), in MIN6 cells treated with 20 mM TEA. (**E-H**) Effect of maximal PKA activity on subcellular CaNAR2 responses. Representative time-courses showing the KCl-induced cytoCaNAR2 response in the (**E**) absence (n = 26) or (**G**) presence (n = 14) of 100 µM IBMX treatment in MIN6 cells, and representative time-courses showing the KCl-induced erCaNAR2 response in the (**F**) absence (n = 18) or (**H**) presence (n = 8) of 100 µM IBMX treatment in MIN6 cells. IBMX was added prior to the initial KCl treatment, and the IBMX concentration in the experiment was maintained by re-addition of 100 µM IBMX after washing out KCl.

The following figure supplement is available for figure 5:

**Figure supplement 1**. Blocking oscillatory PKA activity in the cytosol and at the ER surface in MIN6 cells.

differ subcellularly, with more rapid calcineurin activation occurring in dendritic spines than in the cytosol and nucleus (*Wu et al., 2012*). Calcineurin dynamics are also predicted to differ significantly within the dyadic cleft and cytosol in cardiomyocytes (*Saucerman and Bers, 2008*). In keeping with these findings, our investigation revealed subcellular differences in the temporal pattern of calcineurin activity in response to Ca²⁺ oscillations in pancreatic β-cells. Specifically, cytosolic and plasma membrane calcineurin activity was observed to integrate Ca²⁺ oscillations, whereas Ca²⁺ oscillations evoked intermittent, oscillating calcineurin activity at the ER and mitochondria. Given the wide variety of cellular functions regulated by calcineurin signaling and the significant role of subcellular compartments

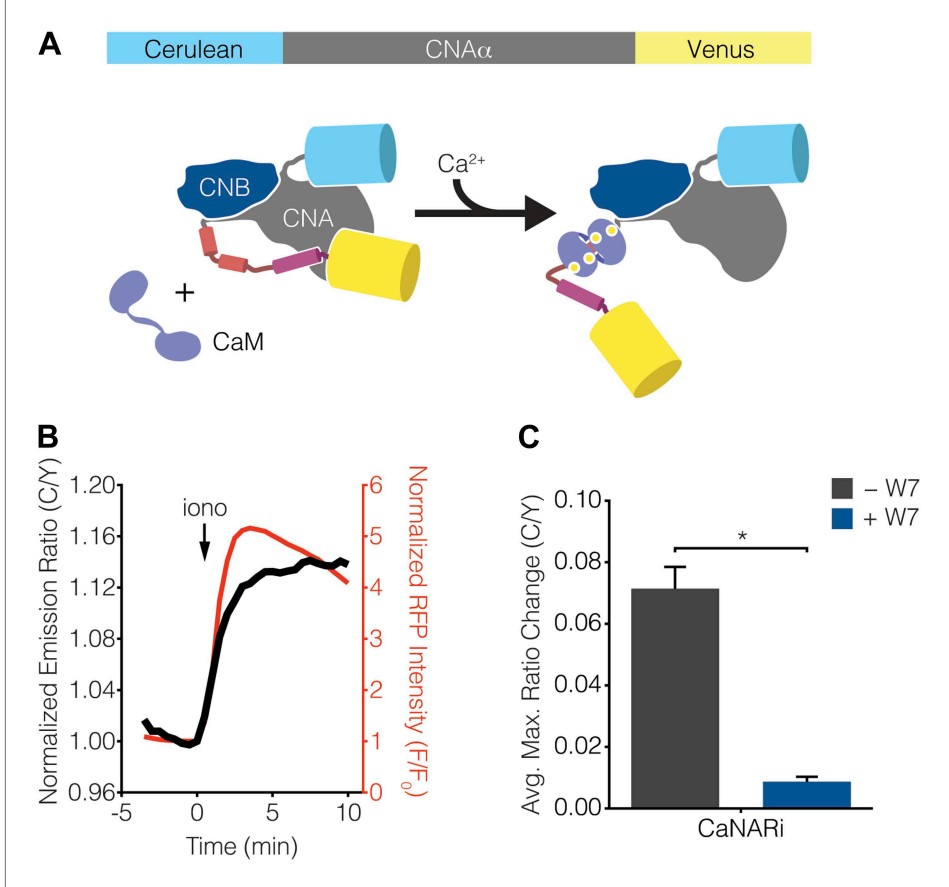

**Figure 6**. Development of CaNARi, a FRET-based reporter of calcineurin activation. (**A**) Schematic illustrating the domain structure of CaNARi and the proposed $Ca^{2+}$/CaM-induced molecular switch. (**B**) Representative time-course showing the cyan/yellow (C/Y) emission ratio change from CaNARi (black curve), along with the RCaMP response (red curve), in HEK293 cells treated with 1 μM ionomycin (iono). (**C**) Summary of C/Y emission ratio responses from CaNARi in HEK293 cells stimulated using 1 μM iono with (+W7, n = 18) or without (−W7, n = 21) pretreating for 30 min with 50 μM of the CaM antagonist W7. Data shown are presented as mean ± SEM. *p < 0.0001.

in modulating signaling molecule behavior (see *Mehta and Zhang, 2010*), this phenomenon is likely to shape calcineurin activity patterns in other cell types as well.

Our investigation into the spatiotemporal dynamics of calcineurin signaling turns on the use of a pair of FRET-based biosensors, each giving distinct responses based on its specific properties and thereby offering a multifaceted view of calcineurin behavior in living cells. The CaNAR family, including CaNAR2 and its precursor CaNAR1 (*Newman and Zhang, 2008*), utilizes the well-characterized dephosphorylation of NFAT to report on the substrate-level dynamics of calcineurin activity and is sensitive to multiple cellular factors, such as both phosphatase and kinase activity. We also generated CaNARi, which reports on the activation of calcineurin upon the binding of $Ca^{2+}$/CaM. The CaNARi response is exclusively determined by the intrinsic affinities between $Ca^{2+}$, CaM, and calcineurin, and CaNARi revealed largely uniform subcellular calcineurin activation patterns during $Ca^{2+}$ oscillations. On the other hand, CaNAR was able to detect clear subcellular variations in calcineurin activity, which stems from the fact that CaNAR monitors the net endogenous calcineurin activity at each location. The enzymatic nature of CaNAR, wherein calcineurin dephosphorylates many probe molecules, also makes it more sensitive for detecting weak calcineurin signals. However, given the somewhat peculiar dephosphorylation and rephosphorylation behavior of NFAT (*Okamura et al., 2000*; *Tomida et al., 2003*), CaNAR may not reflect calcineurin activity towards all targets. Additional approaches, including redesigned CaNARs, are therefore needed to provide a more complete picture.

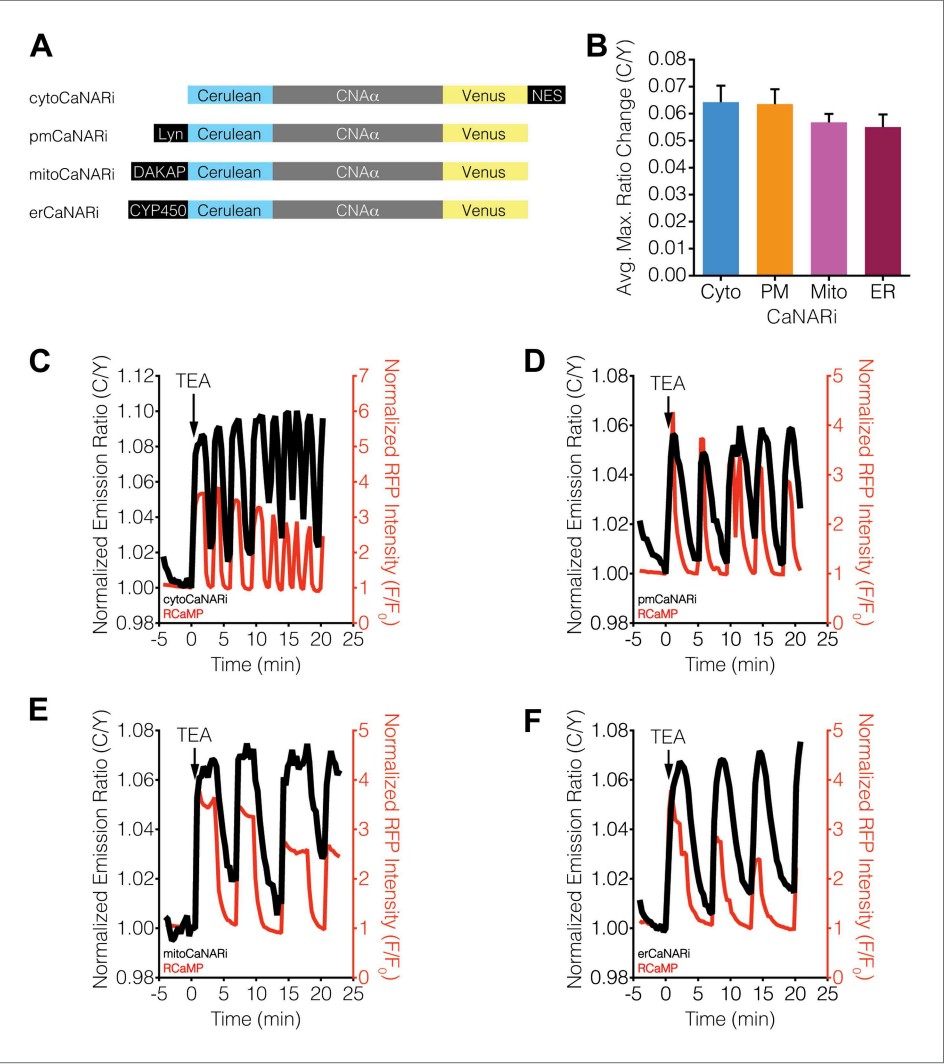

**Figure 7**. Ca²⁺ oscillations induce uniform subcellular calcineurin activation patterns. (**A**) Schematic illustrating the domain structures of the subcellularly targeted CaNARi variants. (**B**) Summary of the C/Y emission ratio responses from each subcellularly targeted CaNARi variant. Data shown are presented as mean ± SEM, with n = 16, 13, 10, and 12 for cytoCaNARi, pmCaNARi, mitoCaNARi, and erCaNARi, respectively. (**C**–**F**) Representative time-courses showing the responses from (**C**) cytoCaNARi, (**D**) pmCaNARi, (**E**) mitoCaNARi, and (**F**) erCaNARi (black curves), along with the response from RCaMP (red curves), in MIN6 cells treated with 20 mM TEA.

The following figure supplement is available for figure 7:

**Figure supplement 1**. Subcellular CaNARi responses are not affected by the reporter expression level.

The differential calcineurin activity dynamics suggest that different cellular compartments are tuned for regulating distinct calcineurin targets. For example, mice specifically lacking calcineurin in β-cells show impaired insulin production and decreased expression of several critical genes, all of which are regulated by the calcineurin-dependent transcription factor NFAT (***Heit et al., 2006***). The efficient activation and nuclear translocation of NFAT in response to Ca²⁺ oscillations requires periods of sustained calcineurin activity to produce a cytoplasmic pool of dephosphorylated NFAT that is available for nuclear import (***Tomida et al., 2003***). Correspondingly, the sustained, integrating calcineurin activity patterns we observed in the cytosolic and plasma membrane regions of MIN6 cells indicated that this signaling domain is optimized for the oscillatory control of transcriptional regulation. In fact, the plasma membrane may play a particularly crucial role in promoting calcineurin/NFAT signaling. The scaffolding protein AKAP79/150 has specifically been shown to recruit calcineurin into a signaling

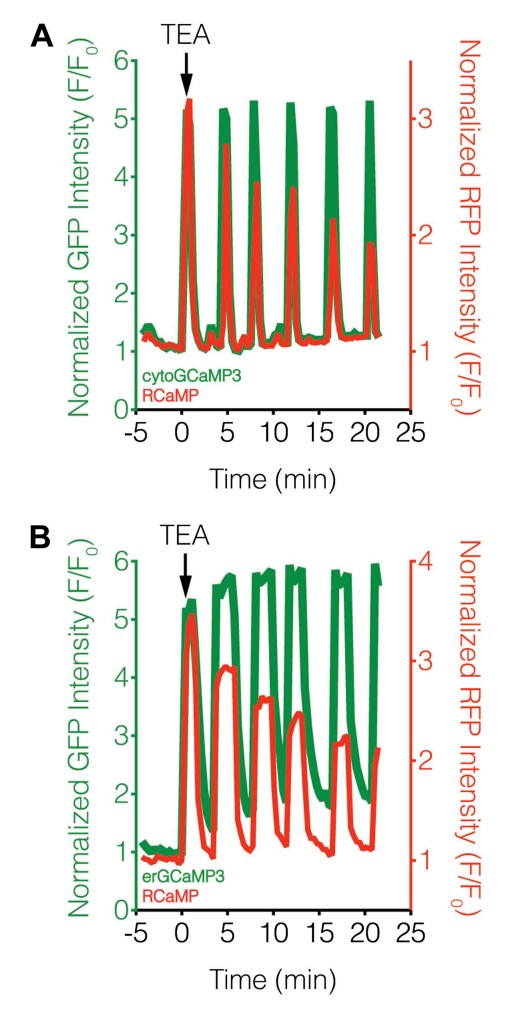

**Figure 8**. Subcellular Ca²⁺ dynamics match global Ca²⁺ dynamics in MIN6 cells. Representative curves showing the response from (**A**) cytoGCaMP3 (n = 23) or (**B**) erGCaMP3 (n = 38) (green curves) and RCaMP (red curves) in MIN6 cells treated with 20 mM TEA. (**A**) To compare the Ca²⁺ dynamics in different subcellular compartments in MIN6 cells, GCaMP3 and RCaMP were first co-expressed in the cytosol. TEA stimulation induced completely overlapping responses from both probes, indicating that they have similar properties, though the RCaMP response amplitude often decreased over time. (**B**) GCaMP3 was then targeted to the ER surface while RCaMP was kept in the cytosol. Upon TEA stimulation, the GCaMP3 response again tracked closely with the RCaMP response, with no noticeable differences in the timing of Ca²⁺ increases or decreases between the ER and cytosol. The basal GCaMP response drifted upwards at the ER surface, but this did not affect the overall dynamics of the Ca²⁺ oscillations.

complex located at the C-terminus of the L-type voltage-gated Ca²⁺ channel (VGCC) in hippocampal neurons (*Oliveria et al., 2007*), and the presence of anchored calcineurin within this complex was required for proper NFAT-dependent nuclear signaling in these cells (*Oliveria et al., 2007*; *Li et al., 2012a*). Both Ca²⁺ influx through the L-type VGCC and AKAP anchoring of calcineurin are also important for insulin secretion (*Bokvist et al., 1995*; *Wiser et al., 1999*; *Barg et al., 2001*; *Lester et al., 2001*; *Hinke et al., 2012*), therefore it is possible that an AKAP-calcineurin-VGCC complex is similarly involved in calcineurin/NFAT signaling in β-cells.

In contrast to the cytosol and plasma membrane, however, the oscillatory calcineurin activity we observed near the ER and mitochondria in MIN6 cells suggests weak and intermittent NFAT activation (*Tomida et al., 2003*). As such, transcriptional signaling via NFAT is likely not the primary function of calcineurin in these subcellular regions. Our previous findings indicate that oscillatory signals can spatially restrict enzyme activity (*Ni et al., 2010*); thus, calcineurin activity oscillations may alternatively be directed towards local ER and mitochondrial targets. For instance, calcineurin was recently shown to dephosphorylate the Ca²⁺-dependent chaperone calnexin and interact with the ER kinase PERK to modulate protein folding and ER stress (*Bollo et al., 2010*), while calcineurin and PERK also jointly regulate insulin secretion (*Wang et al., 2013*). Similarly, calcineurin dephosphorylates and activates the mitochondrial fission protein Drp1 (*Cribbs and Strack, 2007*; *Cereghetti et al., 2008*, *2010*; *Slupe et al., 2013*). Interestingly, Drp1-mediated mitochondrial division involves direct contact between the outer mitochondrial membrane and ER tubules (*Friedman et al., 2011*). It is conceivable that such contact points are hotspots for local calcineurin signaling; the ER and mitochondria are in fact characterized by extensive physical and functional coupling (reviewed in *de Brito and Scorrano, 2010*; *Rowland and Voeltz, 2012*).

Further analyses revealed that PKA, which opposes calcineurin activity towards Drp1 (*Cribbs and Strack, 2007*), antagonizes calcineurin at the ER and helps give rise to calcineurin activity oscillations. Curiously, inhibiting PKA in TEA-stimulated MIN6 cells led to steadily increasing ER calcineurin activity, despite cytosolic Ca²⁺ apparently returning to basal levels. This effect was blocked by calcineurin inhibition and may be due to residual activity from ER Ca²⁺ release channels or to passive Ca²⁺ leak from the ER (*Camello et al., 2002*; *Szlufcik et al., 2012*; *Hammadi et al., 2013*). However, PKA activity was not sufficient to produce calcineurin activity oscillations, as even saturating amounts of PKA activity did not lead to

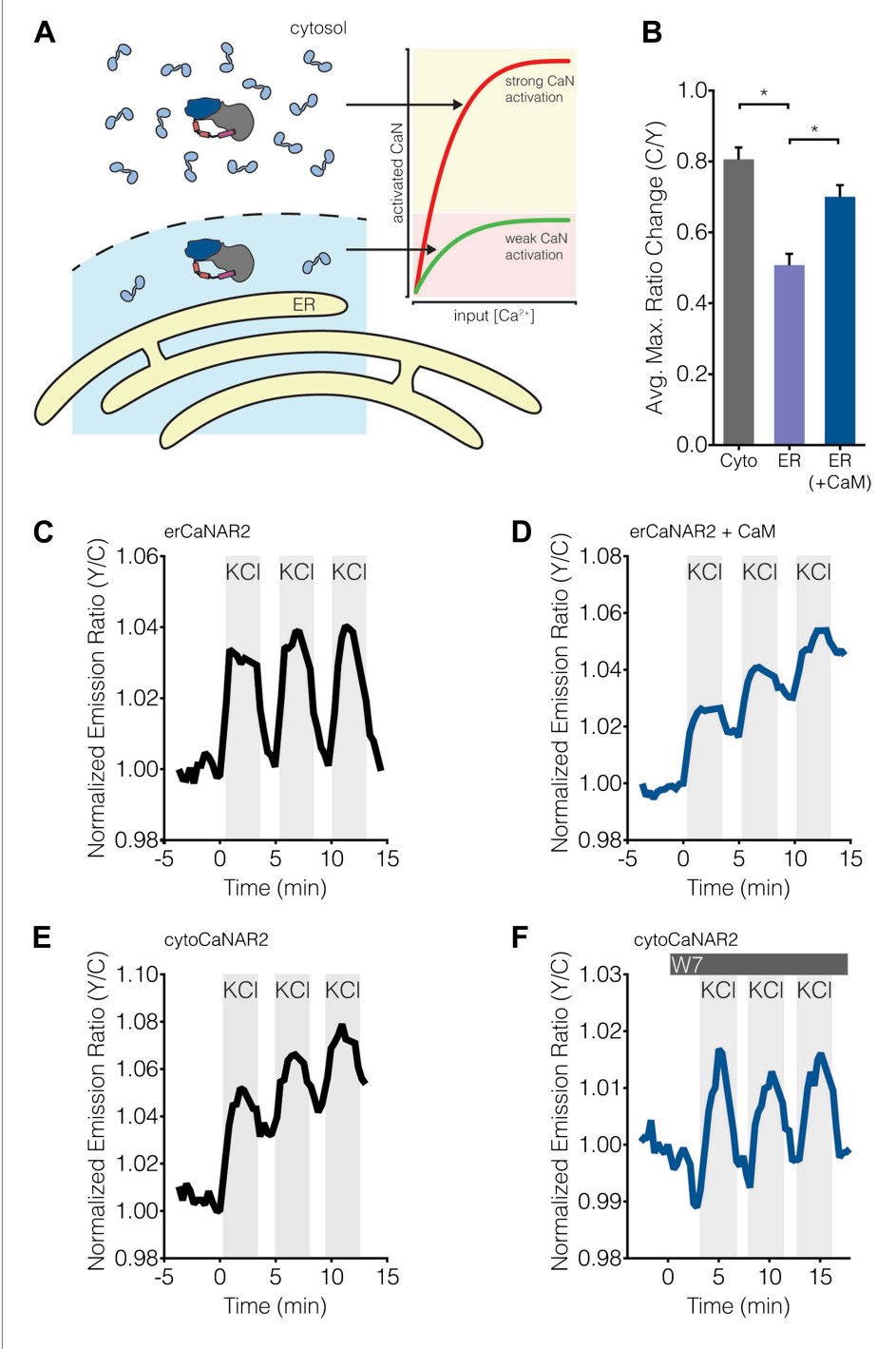

**Figure 9**. Subcellular Ca²⁺/CaM levels determine local calcineurin activity dynamics. (**A**) Model for the regulation of subcellular calcineurin activity in MIN6 cells by local variations in free Ca²⁺/CaM levels. As depicted in this illustration, CaM is predicted to be relatively abundant in the cytosol, thereby leading to strong activation of calcineurin (CaN) in the cytosol (red curve). Conversely, CaM is predicted to be present in much lower quantities near the ER surface, thereby leading to weaker levels of calcineurin activation at any given Ca²⁺ concentration in this part of the cell (green curve). (**B**) Summary of the C/Y emission ratio responses of BSCaM-2 expressed in the cytosol (Cyto) and at the ER surface without (ER) or with (ER+CaM) the overexpression of CaM in MIN6 cells. To achieve maximal levels of free Ca²⁺/CaM, cells were treated with 5 mM CaCl₂ and 5 μM ionomycin. Data shown are presented as mean ± SEM, with n = 31, 33, and 34 for Cyto, ER, and ER+CaM, respectively. *p < 0.0001. (**C** and **D**) Representative

*Figure 9. Continued*

time-courses showing the KCl-induced responses from erCaNAR2 in the (**C**) absence (n = 18) and (**D**) presence (n = 7) of CaM overexpression in MIN6 cells. (**E** and **F**) Representative time-courses showing the KCl-induced responses from cytoCaNAR2 in the (**E**) absence (n = 26) and (**F**) presence (n = 23) of 20 μM W7 treatment in MIN6 cells. W7 was added prior to the initial KCl treatment, and the W7 concentration in the experiment was maintained by re-addition of 20 μM W7 after washing out KCl.

The following figure supplement is available for figure 9:

**Figure supplement 1**. Subcellular BSCaM-2 responses are not affected by the reporter expression level.

CaNAR oscillations in the cytosol. AKAPs may be involved in locally promoting the PKA-mediated antagonism of calcineurin signaling, and DAKAP1 (AKAP121), which binds calcineurin (*Abrenica et al., 2009*), in fact targets to both the mitochondrial and ER membranes (*Ma and Taylor, 2008*). AKAP79 is also known to anchor PKA and calcineurin to the plasma membrane in β-cells (*Lester et al., 2001*), yet calcineurin responses nevertheless appear integrative in this region. We also detected less free $Ca^{2+}$/CaM near the ER surface compared with the cytosol, and CaM overexpression led to integrating rather than oscillating ER calcineurin activity, whereas inhibiting CaM produced the opposite effect in the cytosol. Both of these results are consistent with a model in which limited access to CaM at the ER leads to weak calcineurin activity. Since excess calcineurin activity can lead to β-cell dysfunction and death (*Bernal-Mizrachi et al., 2010*), often involving ER stress and apoptosis, this combination of PKA activity and limited access to CaM may help constrain local calcineurin signals within physiologically permissible limits.

Our results clearly show that the distribution of CaM directly determines subcellular calcineurin activity during β-cell $Ca^{2+}$ oscillations. CaM acts as both a dedicated regulatory subunit and a promiscuous binding partner depending on the specific target (*Saucerman and Bers, 2012*), and whereas promiscuous CaM is free to associate with and dissociate from target proteins as a function of $Ca^{2+}$, dedicated CaM remains bound irrespective of changes in the $Ca^{2+}$ concentration. Subcellular variations in the amounts of dedicated CaM targets may therefore affect the levels of promiscuous CaM that can activate calcineurin in response to $Ca^{2+}$ signals. Moreover, the $Ca^{2+}$-binding kinetics of CaM limit its range of action and potentially require CaM to be activated within $Ca^{2+}$ microdomains (*Saucerman and Bers, 2012*). In β-cells, $Ca^{2+}$ oscillations are often driven by VGCCs in the plasma membrane (*Ashcroft and Rorsman, 1989*; *Hellman et al., 1992*; *Bokvist et al., 1995*; *Mears, 2004*; *Tengholm and Gylfe, 2008*), and CaM has been found to be locally enriched near these channels (*Mori, 2004*), ready to be activated by local $Ca^{2+}$. AKAP79 has also been shown to bind CaM (*Gold et al., 2011*), thereby serving as another potential source for local $Ca^{2+}$/CaM and also perhaps lowering the ability of anchored PKA to antagonize local calcineurin activity. However, additional studies are needed to unravel the precise mechanisms underlying the spatial differences in CaM levels.

In neurons and cardiomyocytes, calcineurin is implicated to behave as an integrator or peak number counter in response to $Ca^{2+}$ oscillations (*Saucerman and Bers, 2008*; *Song et al., 2008*; *Li et al., 2012b*; *Fujii et al., 2013*). In these cells, rapid (i.e., > 1 s$^{-1}$) $Ca^{2+}$ transients, combined with the slow dissociation of calcineurin and $Ca^{2+}$/CaM, facilitate persistent calcineurin activation (*Song et al., 2008*). This also engenders the preferential activation of calcineurin at lower oscillatory frequencies compared with other targets (e.g., CaMKII), resulting in frequency decoding. In β-cells, on the other hand, $Ca^{2+}$ oscillations are slow (i.e., < 1 min$^{-1}$) (*Hellman et al., 1992*; *Tengholm and Gylfe, 2008*), and our results indicate that $Ca^{2+}$/CaM fully dissociates from calcineurin in between each $Ca^{2+}$ peak, suggesting that frequency modulation may not be a prominent feature in β-cell calcineurin signaling. Rather, we found that CaM plays a critical role in spatial signaling under these conditions, which has been hinted at previously (*Teruel et al., 2000*). Furthermore, the frequency decoding observed in neurons and cardiomyocytes is based entirely on the intrinsic kinetics of CaM-target interactions where CaM plays a passive role. Conversely, our analyses highlight a novel mechanism whereby cells utilize CaM to actively encode spatial information, which is then decoded by calcineurin to ensure that oscillatory $Ca^{2+}$ signals are transduced properly within specific local contexts.

## Materials and methods

### CaNAR2 construction

Each CaNAR variant was generated by sandwiching the substrate domain of CaNAR1, which corresponds to amino acids 1–297 from the N-terminus of NFAT1 (*Newman and Zhang, 2008*), between different cyan (CFP) and yellow fluorescent protein (YFP) variants. Cerulean (*Rizzo et al., 2004*), Cerulean2, Cerulean3 (*Markwardt et al., 2011*), circularly permuted Venus (E172) (*Nagai et al., 2004*), and YPet (*Nguyen and Daugherty, 2005*) were each subcloned using *Bam*HI and *Sph*I restriction sites (for CFP) or *Sac*I and *Eco*RI restrictions sites (for YFP) to replace the original ECFP or circularly permuted Venus (L194) in CaNAR1. All CaNAR constructs were generated in the pRSETB vector (Invitrogen, Carlsbad, CA) and then subcloned into pCDNA3 (Invitrogen) for subsequent mammalian expression. Plasma membrane-, outer mitochondrial membrane-, and ER-targeted CaNAR2 constructs were generated by in-frame fusion of full-length CaNAR2 with the N-terminal 11 amino acids from Lyn kinase, the N-terminal 30 amino acids from DAKAP1, and the N-terminal 27 amino acids from CYP450, respectively. Similarly, cytosolic CaNAR2 was generated by in-frame fusion of a C-terminal NES (LPPLERLTL) immediately prior to the stop codon. All constructs were verified by sequencing.

### CaNARi construction

CaNARi was generated in pCDNA3.1(+) (Invitrogen). The full-length α isoform of CNA was PCR amplified from pETCNα (*Mondragon et al., 1997*), a gift of Fan Pan (Johns Hopkins School of Medicine, Baltimore, MD), using primers incorporating a 5′ *Bam*HI site and 3′ *Xho*I site. Cerulean was PCR amplified from plasmid DNA using a forward primer encoding a 5′ *Hind*III restriction site followed by a Kozak consensus sequence for mammalian expression (*Kozak, 1987*) and a reverse primer encoding a 3′ *Bam*HI site. Similarly, Venus was PCR amplified from plasmid DNA using primers encoding a 5′*Xho*I site and 3′ *Xba*I site. The resulting PCR fragments were digested with the corresponding restriction enzymes, gel purified, and ligated into pCDNA3.1(+). To generate the plasma membrane-, outer mitochondrial membrane-, and ER-targeted CaNARi constructs, Cerulean was PCR amplified from plasmid DNA using nested forward primers encoding a 5′ *Hind*III site, a Kozak translation initiation sequence, and the N-terminal targeting sequence from either Lyn kinase, DAKAP1, or CYP450, along with a reverse primer encoding a 3′ *Bam*HI site. The PCR fragments were then subcloned into CaNARi in pCDNA3.1(+) using *Hind*III and *Bam*HI to replace the original Cerulean sequence. Similarly, cytosol-targeted CaNARi was generated by PCR amplification of Venus using a forward primer encoding a 5′ *Xho*I site and nested reverse primers encoding a 3′ *Xba*I site and an NES. The PCR fragment was subcloned into CaNARi in pCDNA3.1(+) using *Xho*I and *Xba*I to replace the original Venus sequence. All constructs were verified by sequencing.

### Other plasmids

ER-targeted AKAR4 was generated from AKAR4 (*Depry et al., 2011*) as above. The Ca²⁺/CaM biosensor BSCaM-2 (*Persechini and Cronk, 1999*; *Tran et al., 2003*) was a gift of Dr Anthony Persechini (University of Missouri–Kansas City, Kansas City, MO). ER-targeted BSCaM-2 was generated by subcloning full-length BSCaM-2 between the *Hind*III and *Xba*I sites of pCDNA3.1(+) containing the N-terminal targeting sequence from CYP450. The CaM-mCherry construct was generated by PCR amplification of full-length CaM (without a stop codon) from plasmid DNA using primers encoding a 5′ *Nhe*I site and a 3′ *Bam*HI site and PCR amplification of mCherry (with a stop codon) from plasmid DNA using primers encoding a 5′ *Bam*HI site and 3′ *Eco*RI site. The resulting PCR fragments were digested using the corresponding restriction enzymes, gel purified, and ligated into pCDNA3.1(+). The red-fluorescent Ca²⁺ sensor RCaMP (*Akerboom et al., 2013*) and the green-fluorescent Ca²⁺ sensor GCaMP3 (*Tian et al., 2009*) were kind gifts of Dr Loren Looger (Janelia Farm Research Campus, HHMI, Ashburn, VA). Subcellularly targeted versions of GCaMP3 were generated by PCR amplification of GCaMP3 (with or without a stop codon) using primers encoding a 5′ *Bam*HI site and a 3′ *Eco*RI site and subcloning into a plasmid containing either an NES or an ER-targeting sequence as above. The high affinity FRET-based Ca²⁺ sensor YC-Nano50 (*Horikawa et al., 2010*) was provided Dr Takeharu Nagai (Hokkaido University, Sapporo, Hokkaido, Japan). All constructs were verified by sequencing.

### Cell culture and transfection

HEK293 cells were cultured in Dulbecco modified Eagle medium (DMEM; Gibco, Grand Island, NY) containing 1 g/l glucose and supplemented with 10% (vol/vol) fetal bovine serum (FBS, Sigma, St. Louis, MO)

and 1% (vol/vol) penicillin-streptomycin (Pen-Strep, Sigma-Aldrich, St. Louis, MO). MIN6 β-cells were cultured in DMEM containing 4.5 g/l glucose and supplemented with 10% (vol/vol) FBS, 1% (vol/vol) Pen-Strep, and 50 μM β-mercaptoethanol. All cells were maintained in a humidified incubator at 37°C with a 5% $CO_2$ atmosphere. Prior to transfection, cells were plated onto sterile, 35-mm glass-bottomed dishes and grown to 50–70% confluence. Cells were then transfected using calcium phosphate and grown an additional 24–48 hr (HEK) or using Lipofectamine 2000 (Invitrogen) and grown an additional 48–96 hr (MIN6) before imaging.

### Reporter localization

MIN6 β-cells expressing either mitochondrial or ER-targeted CaNAR2 were stained for 30 min with MitoTracker RED (Molecular Probes, Eugene, OR) or ER-Tracker RED (Molecular Probes), respectively, at a final concentration of 1 μM in Hank's Balanced Salt Solution (HBSS). These cells, as well as cells expressing cytosolic or plasma membrane-targeted CaNAR2, were imaged on a Nikon Eclipse Ti inverted fluorescence microscope (Nikon Instruments, Melville, NY) equipped with a perfect focus system (Nikon), a 100x/1.49 NA oil-immersion objective lens, an electron multiplying charge coupled device camera (Photometrics, Tucson, AZ), and a laser TIRF system (Nikon). YFP and RFP images were acquired using a 514-nm argon laser (Melles Griot, Rochester, NY) and a 561-nm Sapphire solid-state laser (Coherent, Santa Clara, CA), respectively. The system was controlled using the NIS-Elements software package (Nikon). Exposure times were between 50 and 200 ms. Images were analyzed using ImageJ software (http://imagej.nih.gov/ij/).

### FRET imaging

Cells were washed twice with HBSS and subsequently imaged in the dark at 37°C. Tetraethylammonium chloride (TEA; Sigma), thapsigargin (TG; Sigma), ionomycin (iono; Calbiochem, San Diego, CA), calcium chloride ($CaCl_2$; Sigma), W-7 (Cayman Chemical, Ann Arbor, MI), potassium chloride (KCl; JT Baker, Phillipsburg, NJ), H89 (Sigma), Gö6983 (Sigma), forskolin (Fsk; Calbiochem), and 3-isobutyl-1-methylxanthine (IBMX; Sigma) were added as indicated. Images were acquired using an Axiovert 200M inverted fluorescence microscope (Carl Zeiss, Thornwood, NY) with a 40x/1.3 NA oil-immersion objective lens and a cooled charge-coupled device camera (Roper Scientific, Trenton, NJ) controlled by Metafluor 7.7 software (Molecular Devices, Sunnyvale, CA). Dual cyan/yellow emission ratio imaging was performed using a 420DF20 excitation filter, a 450DRLP dichroic mirror, and two emission filters (475DF40 for CFP and 535DF25 for YFP). RFP intensity was imaged using a 568DF55 excitation filter, a 600DRLP dichroic mirror, and a 653DF95 emission filter. Filter sets were alternated using a Lambda 10–2 filter changer (Sutter Instruments, Novato, CA). Exposure times were between 10 and 500 ms, and images were taken every 20–30 s. Raw fluorescence images were corrected by subtracting the background fluorescence intensity of a cell-free region from the emission intensities of biosensor-expressing cells. Emission ratios (yellow/cyan or cyan/yellow) were then calculated at each time point. All time-courses were normalized by dividing the emission ratio or, in the case of RCaMP, the intensity at each time point by the basal value immediately preceding drug addition.

## Acknowledgements

We wish to thank Loren Looger, Fan Pan, Takeharu Nagai, and Anthony Persechini for generously providing plasmids used in this study. We are also grateful to Fabian Hertel for assistance with the Nikon Eclipse microscope.

## Additional information

### Funding

| Funder | Grant reference number | Author |
|---|---|---|
| National Institutes of Health | R01DK073368 | Jin Zhang |
| National Institutes of Health | NIH Director's Pioneer Award, DP1 CA174423 | Jin Zhang |

The funder had no role in study design, data collection and interpretation, or the decision to submit the work for publication.

## Author contributions

SM, Conceived project and designed experiments; acquired and analyzed data; wrote/revised the manuscript; N-NA-H, Acquired and analyzed data; AG, Acquired and analyzed data; contributed reagents; LO, KG, Contributed reagents; JZ, Conceived project and designed experiments; wrote/revised the manuscript

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
