## [Decision Letter]

[Editors’ note: an earlier version of this submission was rejected after peer review but the authors submitted for further consideration.]

Thank you for choosing to send your work entitled “Calmodulin-controlled spatial decoding of oscillatory Ca^2+^ signals by calcineurin” for consideration at *eLife*. Your full submission has been evaluated by a Senior editor and 2 peer reviewers, one of whom is a member of our Board of Reviewing Editors, and the decision was reached after discussions between the reviewers. We regret to inform you that your work will not be considered further for publication at this time.

The research is novel, with sophisticated use of FRET reporters. The finding that calcineurin activity is restricted in the vicinity of the ER/mitochondria is interesting and unexpected. However, there were a number of substantive concerns raised by the reviewers that mean that substantial revision, including new experiments, would be required before further consideration. These concerns are detailed below. Regrettably, *eLife* policy is to reject papers requiring major revisions.

*Reviewer 1*:

The authors show that NFAT phosphorylation state integrates calcium waves in the cytosol but oscillates with calcium waves at the ER (measured with a calmodulin reporter). Oscillation is expected if each wave of calcium-dependent dephosphorylation (by calcineurin) is fully reversed (by PKA) during the ensuing low calcium period, while integration is expected if reversal is incomplete. Differential NFAT phosphorylation kinetics might be explained if PKA were less active in the cytosol than ER. However, the authors found that PKA activity oscillations are of similar magnitude at the ER and cytosol, so this explanation is unlikely. Alternatively, less calcineurin may be activated near the ER, so it may be more readily antagonized by PKA. Calmodulin is known to be much less abundant than its effectors. To test whether calmodulin is limiting for calcineurin activation near the ER, they over-expressed calmodulin and found that this switched ER calcineurin activity from oscillating to integrating modes. Based on this, the authors conclude that calmodulin competition explains the observations. Another finding is that, even though NFAT phosphorylation kinetics are different in cytosol and at ER, calcineurin conformation changes, which require calcium-calmodulin, oscillate in both localizations.

1) Given that calmodulin is limiting, over-expressing calmodulin-binding reporters will likely upset the normal balance. How does CaNAR2 level compare with endogenous NFAT? If CaNAR2 is more highly expressed at ER than cytosol, that could explain the difference. Similarly, the conformation detector, CaNARi, could be expressed at lower level. Expression levels need to be measured and confirmed to be equivalent.

2) Roles for calcium stores in the ER should be excluded, e.g, using appropriate drugs.

3) The co-oscillation of PKA and calcium is not unexpected, but hinders clear interpretation of the data. The results would be more compelling if PKA activity was buffered to prevent oscillation.

4) Effect of over-expressing calmodulin should be complemented by inhibiting calmodulin. Low doses of W7 or calmodulin siRNA should cause oscillations of NFAT phosphorylation in the cytosol as well as the ER.

5) It is not clear from the model why the calcineurin conformation sensor oscillates at the ER. The magnitude of FRET oscillation is similar at ER and cytosol in the traces shown. If calmodulin were limiting, less magnitude would be expected. The explanation is unclear and needs to be resolved.

*Reviewer 2*:

The work of Zhang et al attempts to analyze spatial differences in calcium oscillation profiles with intact beta cells. The authors chose pancreatic beta cells likely because of their known ability to produce massive calcium oscillations, although the connection to insulin secretion was not investigated in this work. The authors employ improved and locally acting sensors (CaNAR2 and CaNARi) to measure calcineurin activity, BSCaM-2 to sense Ca/CaM levels, and an intensiometric genetically encoded calcium sensor. One of the major findings in the present work is that calcineurin oscillations distant from ER membranes are less dynamic ('integrating') than those close to the ER.

1) I don't understand why the mitochondria region differs from the cytosol response.

2) I also wonder if the calcium oscillations close to the plasma or the ER membrane, respectively, differ in a similar way as the calcineurin responses. Strong support comes from experiments where CaM is overexpressed and the ER-located sensor mimics the cytosol and plasma membrane location. What is missing is the alternative experiment, where calcineurin should be blocked by modest levels of W-7. This should lead to perfect oscillations of calcineurin everywhere.

The difference in the oscillation patterns could also be driven by activities in the respective membranes such as calcium influx close to the plasma membrane or more efficient calcium uptake by SERCA close to the ER. These possibilities have not been investigated and are not discussed.

3) Similarly, effects of the PKA and PKC inhibitors might strongly influence calcium channels including the IP3 receptor and might therefore change the threshold calcium level required for successful oscillations. The hypothesis that there is calcium leakage close to the ER is a weak explanation for the steady increase in calcineurin activity after Goe6983 and H89.

4) The model presented is based on the fact that Ca/CaM activates calcineurin. However, it was previously shown that activation of non-calcium-dependent phosphatases such as PP1 suffice to induce calcium oscillations in the absence of depolarization, likely by changing the phosphorylation level of the IP3 receptor (Reither et al., 2013). Is calcineurin activation also inducing calcium signals? If this were the case, the model would need to be much more complicated.

5) I trust the statement that CaM might be scarce in some cells but how abundant is it in Min6 cells? At least an estimate from the transcriptome should have been provided as the conclusion on subcellular differences in Ca/CaM is based on the scarcity assumption. Further, if the CaM concentration were indeed in the sub 100 nM range, then overexpression of a CaM binding protein (usually in the micromolar range) will definitely buffer CaM close to its location on the ER membrane. There is a chance that either there is enough CaM or if not that the sensor produces an artifact. I fear the authors can't win on this one. Was BSCaM-2 also attached to the plasma membrane and used to determine the CaM concentration there?

In conclusion, although very interesting, there are too many open questions and too many experiments that needed to be done to accept this paper.

---

## [Author Response]

A number of changes, including several new experiments, have been incorporated into the new submission, and we believe that these changes fully address the concerns raised by the reviewers.

Reviewer 1:

*1) Given that calmodulin is limiting, over-expressing calmodulin-binding reporters will likely upset the normal balance. How does CaNAR2 level compare with endogenous NFAT? If CaNAR2 is more highly expressed at ER than cytosol, that could explain the difference. Similarly, the conformation detector, CaNARi, could be expressed at lower level. Expression levels need to be measured and confirmed to be equivalent*.

The reviewer raises an important point regarding the biosensor expression levels. Unfortunately, after searching the literature and consulting with outside experts, we were unable to find any concrete data on the endogenous expression levels of the various NFAT isoforms. Nevertheless, we used purified YFP to calibrate the intensity counts generated on our microscope, and based on this calibration, we were able to estimate that the intracellular concentration of cytoCaNAR2 ranged between approximately 45 nM and 4 μM in our experiments. Importantly, we also found that the response patterns reported by each subcellularly targeted CaNAR2 variant held steady over similar ranges of biosensor expression levels, and the Results section has been updated to reflect this finding (see Figure 2—figure supplement 1). Similarly, we found no meaningful correlation between the expression levels of the calcineurin conformation sensor (CaNARi) or the Ca^2+^/CaM probe (BSCaM-2) and their observed FRET ratio changes. Based on these data, we conclude that biosensor expression did not influence our reported observations.

*2) Roles for calcium stores in the ER should be excluded, e.g. using appropriate drugs*.

We investigated this using subcellularly targeted versions of GCaMP3 co-transfected along with diffusible RCaMP, and we observed no differences between local Ca^2+^ dynamics in the cytosol, at the ER surface, or even at the plasma membrane compared with global Ca^2+^ dynamics. The ER and cytosol data have been added to the revised manuscript. Furthermore, although ER-targeted GCaMP3 showed that basal Ca^2+^ levels near the ER drift upwards during oscillations, this basal drift does not impact the erCaNAR response, suggesting that ER stores are not playing a significant role in the differential calcineurin activities. We did detect a minor difference in the Ca^2+^ amplitude at the ER surface versus the cytosol, and the manuscript has been updated to reflect this observation; however, the fact that altering calmodulin alone, either via overexpression or by W7 inhibition, can change the calcineurin activity patterns points to a more direct role for calmodulin.

*3) The co-oscillation of PKA and calcium is not unexpected, but hinders clear interpretation of the data. The results would be more compelling if PKA activity was buffered to prevent oscillation*.

As suggested by the reviewer, we have updated Figure 5 with experiments in which the PKA activity was clamped at a steady level. To do so, we pretreated cells with the pan-phosphodiesterase inhibitor IBMX, which led to maximal PKA activity, and then treated these cells with repeated doses of KCl followed by wash out. We observed no differences in the CaNAR response patterns under these conditions compared with conditions in which PKA activity was allowed to oscillate – i.e., cytoCaNAR2 had an integrating response and erCaNAR2 had an oscillating response.

*4) Effect of over-expressing calmodulin should be complemented by inhibiting calmodulin. Low doses of W7 or calmodulin siRNA should cause oscillations of NFAT phosphorylation in the cytosol as well as the ER*.

We thank the reviewer for suggesting this experiment. We indeed found that pretreating MIN6 cells with a submaximal dose (20 μM) of W7, followed by repeated doses of KCl, produces completely oscillatory responses from cytoCaNAR2. This new data has been added to Figure 9 in the revised manuscript.

*5) It is not clear from the model why the calcineurin conformation sensor oscillates at the ER. The magnitude of FRET oscillation is similar at ER and cytosol in the traces shown. If calmodulin were limiting, less magnitude would be expected. The explanation is unclear and needs to be resolved*.

Given that the activation, and hence the conformation, of calcineurin is strictly controlled by cytosolic Ca^2+^ elevations, Ca^2+^ oscillations are still expected to induce oscillatory calcineurin activation even if less calcineurin is being activated. In addition, because the FRET response from this sensor is small, it may not be able to reliably report differences in calcineurin activation amplitude. Nevertheless, based on this comment, we compared the response amplitudes more closely and found that the responses from mitochondrial and ER-targeted CaNARi were in fact smaller on average than those from cytosolic and plasma membrane CaNARi. These data are now summarized in Figure 7 of the revised manuscript. Although the difference was not statistically significant, the observed trend supports our model, and we thank the reviewer for prompting this closer look at our data.

Reviewer 2:

*1) I don't understand why the mitochondria region differs from the cytosol response*.

We have chosen to focus on the different activity dynamics between the ER surface and the cytosol. The activity dynamics on the outer mitochondrial membrane were similar to what was observed at the ER, and the underlying mechanism(s) will be examined in future studies.

*2) I also wonder if the calcium oscillations close to the plasma or the ER membrane, respectively, differ in a similar way as the calcineurin responses. Strong support comes from experiments where CaM is overexpressed and the ER-located sensor mimics the cytosol and plasma membrane location. What is missing is the alternative experiment, where calcineurin should be blocked by modest levels of W-7. This should lead to perfect oscillations of calcineurin everywhere*.

*The difference in the oscillation patterns could also be driven by activities in the respective membranes such as calcium influx close to the plasma membrane or more efficient calcium uptake by SERCA close to the ER. These possibilities have not been investigated and are not discussed*.

As mentioned above in response to reviewer 1, we observed no differences between the Ca^2+^ oscillations in the cytosol, at the ER surface, or at the plasma membrane (measured using subcellularly targeted GCaMP3) and global Ca^2+^ oscillations (measured concurrently using diffusible RCaMP), suggesting that local Ca^2+^ signals are not the determining factor for the differential calcineurin responses. The GCaMP/RCaMP curves for the cytosol and ER have been added to the revised manuscript (Figure 8).

We thank the reviewer for suggesting this experiment. As mentioned above, we indeed found that pretreating MIN6 cells with a submaximal dose (20 μM) of W7, followed by repeatedly adding and washing out KCl, produces completely oscillatory responses from cytoCaNAR2. These new data have been added to Figure 9 in the revised manuscript.

*3) Similarly, effects of the PKA and PKC inhibitors might strongly influence calcium channels including the IP3 receptor and might therefore change the threshold calcium level required for successful oscillations. The hypothesis that there is calcium leakage close to the ER is a weak explanation for the steady increase in calcineurin activity after Goe6983 and H89*.

The reviewer is correct that the inhibition of PKA or PKC could alter the behavior of Ca^2+^ channels. For example, PKA has been shown to stimulate the activity of both the L-type VGCC and the IP3 receptor; thus, inhibiting PKA should lead to decreased Ca^2+^ influx, which is in fact what we observe in Figure 3. Conversely, PKC has been shown to inhibit both of these channels, and we also see in Figure 3 that inhibiting PKC leads to increased Ca^2+^ levels and, potentially, increased calcineurin activity (see responses to reviewer 1 above). Whereas the increase in calcineurin activity after PKC inhibition is clearly due to the elevated Ca^2+^, we attributed the increased calcineurin activity after PKA inhibition to Ca^2+^ leak because of the decreased Ca^2+^ level and the likely decrease in IP3R-mediated Ca^2+^ release, though the latter may still be having an effect.

*4) The model presented is based on the fact that Ca/CaM activates calcineurin. However, it was previously shown that activation of non-calcium-dependent phosphatases such as PP1 suffice to induce calcium oscillations in the absence of depolarization, likely by changing the phosphorylation level of the IP3 receptor (Reither et al., 2013). Is calcineurin activation also inducing calcium signals? If this were the case, the model would need to be much more complicated*.

Calcineurin has been shown to interact with ER Ca^2+^ release channels, such as the IP3 receptor, and may potentially modulate their activity, though this remains somewhat controversial. In addition, calcineurin is known to inhibit influx through plasma membrane Ca^2+^ channels. Therefore, it is more likely that calcineurin is modulating existing Ca^2+^ signals rather than inducing them per se. It is possible that calcineurin exerts other effects to stimulate Ca^2+^ influx; however, there is currently no firm evidence that would lead us to pursue this direction.

*5) I trust the statement that CaM might be scarce in some cells but how abundant is it in Min6 cells? At least an estimate from the transcriptome should have been provided as the conclusion on subcellular differences in Ca/CaM is based on the scarcity assumption. Further, if the CaM concentration were indeed in the sub 100 nM range, then overexpression of a CaM binding protein (usually in the micromolar range) will definitely buffer CaM close to its location on the ER membrane. There is a chance that either there is enough CaM or if not that the sensor produces an artifact. I fear the authors can't win on this one. Was BSCaM-2 also attached to the plasma membrane and used to determine the CaM concentration there? In conclusion, although very interesting, there are too many open questions and too many experiments that needed to be done to accept this paper*.

We have not used this BSCaM-2-based approach to determine free Ca^2+^/CaM concentrations; rather, our aim was simply to compare the relative Ca^2+^/CaM levels in different compartments, as has been previously done using these probes (92). To our knowledge, this type of information is difficult to obtain using other methods. Both cytosolic and plasma membrane-targeted BSCaM-2 showed identical responses (80.61 ± 0.03% vs. 79.15 ± 0.04% FRET ratio change), whereas the response at the ER surface is lower (50.74 ± 0.03%), suggesting that the free Ca^2+^/CaM level is lower in this compartment. No downstream signaling processes were monitored in the BSCaM-2 experiments, thereby limiting the potential for buffering effects, and we also found no correlation between probe expression and the FRET response over a large range of expression levels, which we would have expected to see were the biosensor levels impacting the results. Furthermore, overexpressing CaM or inhibiting CaM using W7 clearly modulated the ER and cytosolic CaNAR responses, respectively, not to mention the fact that overexpressing CaM rescued the BSCaM-2 response at the ER surface (70.03 ± 0.03%). These observations in particular strongly suggest that free Ca^2+^/CaM levels are limiting near the ER surface, resulting in weaker calcineurin activation at this location. Similar observations were made by Teruel and colleagues, who reported smaller increases in free nuclear Ca^2+^/CaM levels compared with the cytosol in response to Ca^2+^ transients (92). We have now cited their study in the manuscript.